# Deep Shells: Unsupervised Shape Correspondence with Optimal Transport

**Marvin Eisenberger**
Technical University of Munich
`marvin.eisenberger@in.tum.de`

**Aysim Toker**
Technical University of Munich
`aysim.toker@in.tum.de`

**Laura Leal-Taixé**
Technical University of Munich
`leal.taixe@tum.de`

**Daniel Cremers**
Technical University of Munich
`cremers@tum.de`

## Abstract

We propose a novel unsupervised learning approach to 3D shape correspondence that builds a multiscale matching pipeline into a deep neural network. This approach is based on smooth shells, the current state-of-the-art axiomatic correspondence method, which requires an a priori stochastic search over the space of initial poses. Our goal is to replace this costly preprocessing step by directly learning good initializations from the input surfaces. To that end, we systematically derive a fully differentiable, hierarchical matching pipeline from entropy regularized optimal transport. This allows us to combine it with a local feature extractor based on smooth, truncated spectral convolution filters. Finally, we show that the proposed unsupervised method significantly improves over the state-of-the-art on multiple datasets, even in comparison to the most recent supervised methods. Moreover, we demonstrate compelling generalization results by applying our learned filters to examples that significantly deviate from the training set.

## 1 Introduction

Computing shape correspondence is a fundamental component in understanding the 3D world, and is at the heart of many applications in computer vision and graphics. It is also a notoriously hard problem and a multitude of different solutions have been proposed over the years. Classical axiomatic methods make assumptions about the geometric properties of the input surfaces, like small local distortions, to compute correspondences for a well-defined class of objects. Many such methods rely on hand-crafted features [40, 39, 3] and most of them make restrictive assumptions about the discretization, topology or morphology of the considered objects. A promising venture for extending shape matching methods to a broader range of inputs is to apply machine learning to geometric data. While deep learning has achieved great success in the field of image analysis, extending the power of deep networks to non-Euclidean data remains an important and very actively studied open challenge. Although there has been much progress, many methods to date lack a strong geometric foundation that fully acknowledges the underlying structure of 3D surfaces. A recent line of research pioneered by [20] aims at combining a learnable local feature extractor with the axiomatic approach functional maps [26]. This combination of machine learning with axiomatic methods is an important breakthrough since it allows us to integrate geometric priors into our model which can significantly enhance the learning process. Most follow up work inspired by deep functional maps [20] use functional maps as a matching layer. The underlying assumption of these approaches is that the input pairs are nearly isometric. This manifests in energy functions that try to preserve intrinsic quantities like the Laplace-Beltrami operator [34, 10] or geodesic distances [14]. As a consequence,

current state-of-the-art methods require expensive postprocessing via an axiomatic method and they typically do not generalize well to previously unseen data and non-isometric pairs. Instead of the purely intrinsic functional maps method, we build upon more recent advances in extrinsic-intrinsic axiomatic shape matching [11].

**Contribution**  In this work, we systematically derive a deepified version of the current state-of-the-art axiomatic shape correspondence method smooth shells [11] from optimal transport. This allows us to integrate it into an end-to-end trainable deep network and learn optimal local features. Most prior work improves hand-crafted input features independently per vertex using shared weights. We show that this approach tends to be unstable for imperfect inputs. Instead, we use a manifold CNN architecture based on smooth spectral filters [7, 15]. Our geometric loss function measures the alignment tightness of the obtained matching which can be computed without any supervision. Finally, we show quantitative results that compare favorably to the state-of-the-art, both in terms of axiomatic and machine learning methods. In comparison to closely related learning approaches, the strong geometric nature of our method yields high-quality correspondences without the need for expensive pre or postprocessing.

## 2 Background

### 2.1 Manifold learning

The unprecedented success of convolutional neural networks (CNNs) on tasks like image and natural language processing suggests that there is a big potential of devising similar architectures for non-Euclidean data. Here, we give a non-exhaustive account of these geometric deep learning techniques on Riemannian manifolds $\mathcal{X}$ that directly imitate CNNs. For a more detailed review, we refer the reader to [6]. One straightforward approach is to learn high-level features as local correlation patterns which are explicitly defined as intrinsic patch operators [22, 5, 25, 29, 35]. While these charting based techniques have proven to be useful for 3D shapes, they make rather strong assumptions about the structure of the data and are therefore not trivially transferable to different domains like e.g. graphs or general manifolds. A complementary approach is to compute a convolution of signals $F : \mathcal{X} \to \mathbb{R}^L$ with some filter in the spectral space where it is simply a pointwise multiplication with learnable diagonal coefficient matrices $\Gamma_{l',l}$ [7]:

$$G_{l'} = h\left(\sum_{l=1}^{L} \Phi\Gamma_{l',l}\Phi^{\dagger}F_l\right). \tag{1}$$

This indeed corresponds to a convolution on the manifold $\mathcal{X}$ because of the well-known property that a convolution is a linear operator that commutes with the Laplacian. In this context, $\Phi$ is the spectral basis and $h$ is some non-linearity. In general, these filters are not transferable between two different surfaces because the Laplacian eigenbasis is domain-dependent. However, it was shown in [15] that using coefficients that are smooth in the frequency domain leads to spatially localized filters:

$$\mathrm{diag}(\Gamma_{l',l}) = B\gamma_{l',l}. \tag{2}$$

Here, $B = \big(b_j(\lambda_i)\big)_{ij}$ is an alternant matrix with smooth basis functions $b_j$ applied to the Laplacian eigenvalues $\lambda_i$ and $\gamma_{l',l}$ are the learnable weights. Note that the idea of using such filters is also the foundation of graph neural networks. In this particular case, the $b_j$'s are polynomials which alleviates the need for computing the spectral embedding of the input signals $F$ explicitly [9, 19]. This construction vastly popularised graph neural networks, see [43] for a recent survey of subsequent work on this topic. Unfortunately, the message passing scheme from GNN's is not suitable for the kind of data that we consider in this work. A continuous 3D surface is typically discretized by a triangular mesh or a point cloud. In contrast to GNN's however, we want to learn local features that are independent of this discretization. Therefore, we will consider more general smooth spectral convolution filters in this work.

### 2.2 Shape correspondence

**Axiomatic methods**  Traditional approaches directly compute a matching $\mathcal{P} : \mathcal{X} \to \mathcal{Y}$ between two input surfaces $\mathcal{X}$ and $\mathcal{Y}$ by making use of certain geometric properties and invariances of deformable

shapes. This is an extensively studied topic and we only focus on approaches that are directly related to ours. For a more complete overview, we refer the reader to surveys on shape correspondence [37, 41] with [36] being the most recent one.

The functional maps [26] framework generalizes the classical correspondence problem by looking for a functional $\mathcal{C} : L^2(\mathcal{X}) \to L^2(\mathcal{Y})$ that maps not points, but functions from $\mathcal{X}$ to $\mathcal{Y}$. The matching $\mathcal{P}(x) = y$ can be recovered from $\mathcal{C}$ by mapping delta distributions $\mathcal{C}(\delta_x) = \delta_y$. In practice, we can opt for a compact representation of functions in a finite basis $(\phi_i)_{1 \leq i \leq K}$ which allows us to represent a functional map as a matrix $\mathbf{C} \in \mathbb{R}^{K \times K}$. The most common choice of basis functions $\phi_i$ are the Laplace-Beltrami eigenfunctions which are provably optimal for representing smooth functions on a manifold [27]. Furthermore, this choice allows us to build certain assumptions about the input objects, like near-isometry or area preservation, directly into the representation [26]. The original framework has been extended to handle partial shapes [32, 21], compute refined point-to-point maps [33], to allow for orientation preserving maps [31] and to iteratively upsample a coarse map [23].

Most recently, [11] proposed smooth shells which combines functional maps with extrinsic shape alignment. This method is able to compute high quality correspondences for challenging pairs but it also requires a good initialization to find a meaningful local minimum. The authors solve this issue by performing a stochastic Markov chain Monte Carlo (MCMC) search over the space of initial poses prior to the main pipeline. While this was shown to work well, it is also costly: In practice, roughly 100 test runs have to be performed to assess the quality of different proposal initializations.

**Machine learning methods**   In Section 2.1, we discussed different manifold learning techniques, many of which have been successfully applied to 3D shape correspondence [22, 5, 25, 29, 35]. A complementary, more task-driven approach is to use a modified axiomatic method where the hand-crafted inputs are replaced by learned features. This line of research was pioneered by deep functional maps [20] which takes SHOT [40] features as an input and refines them independently for each point using a fully connected neural network with shared weights. The resulting learned features are then used as inputs to the functional maps method [26]. Overall, this yields an end-to-end trainable pipeline because functional maps with soft correspondences is differentiable. Further extensions have successfully coupled this framework with an unsupervised loss [14, 34] or with point cloud learning techniques [10]. One thing that all of these methods have in common is that they are using the functional maps pipeline as a matching layer. Moreover, all but the last one use the same network architecture like the original deep functional maps [20] which computes features independently for each point. Similar to our approach, [38] and [44] use a differentiable Sinkhorn layer for image matching and rigid point cloud registration respectively. A more specialized approach to compute correspondences for a specific class of 3D shapes was proposed by [13]. The main idea here is to learn how to deform a template of a specific class of objects, e.g. a human template. To that end, the authors use a pointnet spatial encoder [30] and a decoder in some latent deformation parameterization.

## 3   Deep shells

### 3.1   Product space embedding

Smooth shells [11] is a hierarchical matching method that operates on the product space of extrinsic coordinates and intrinsic features. In particular, it embeds a given input shape $\mathcal{X}$ into $\mathbb{R}^{K+6}$ using the following coordinate function:

$$\mathbf{X}_k := \left( \Phi_k, X_k, \mathbf{n}_k^{\mathcal{X}} \right) : \mathcal{X} \to \mathbb{R}^{K+6}. \tag{3}$$

The intrinsic features $\Phi_k$ are defined as the first $k$ Laplace-Beltrami eigenfunctions, $\mathbf{n}_k^{\mathcal{X}}$ are the outer normals and $X_k$ the smoothed extrinsic coordinates, see [11] for more details. In this embedding space, an alignment between two shapes $\mathcal{X}$ and $\mathcal{Y}$ can be computed by appropriately parameterizing the intrinsic deformation with a functional map [26] and the extrinsic deformation as a pointwise translation in the Laplace-Beltrami eigenbasis $\Phi_k$. Overall, the following energy is minimized:

$$E(\mathcal{P}, \mathbf{C}, \tau) := \left\| \mathbf{X}_k^* - \mathbf{Y}_k \circ \mathcal{P} \right\|_{L_2}^2 = \left\| (\Phi_k \mathbf{C}^\dagger, X_k + \Phi_k \tau, \overset{*}{\mathbf{n}}_k^{\mathcal{X}}) - \mathbf{Y}_k \circ \mathcal{P} \right\|_{L_2}^2. \tag{4}$$

Here, we need to jointly optimize for the unknown correspondences $\mathcal{P} : \mathcal{X} \to \mathcal{Y}$ and the deformation coefficients $\mathbf{C} \in \mathbb{R}^{k \times k}$ and $\tau \in \mathbb{R}^{k \times 3}$. To handle this coupled problem, the authors in [11] define an iterative scheme that gradually increases $k$ with a fixed number of iterations and alternates between optimizing for $\mathcal{P}$ and $(\mathbf{C}, \tau)$.

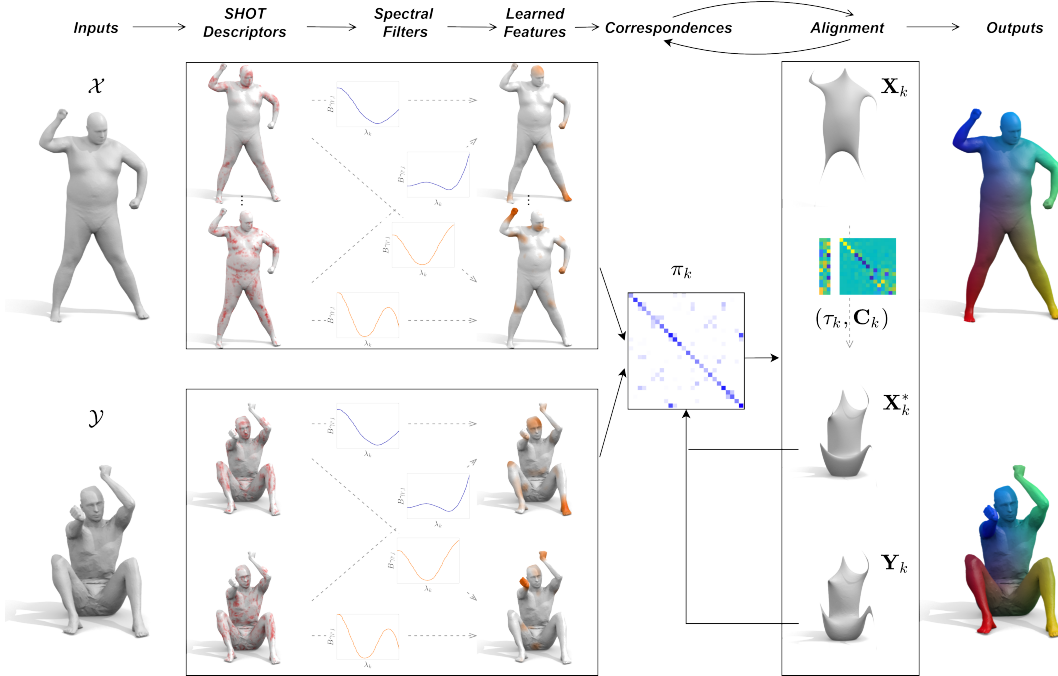

Figure 1: An overview of our network. We compute SHOT [40] descriptors on the two input shapes $\mathcal{X}$ and $\mathcal{Y}$ and apply learnable spectral convolution filters to them to obtain refined local features $G^{\mathcal{X}}$ and $G^{\mathcal{Y}}$. Those are then used to initialize the main matching layer, see Eq. (13). This scheme starts with a coarse approximation of the input geometries $\mathbf{X}_k, \mathbf{Y}_k$ and gradually increases $k$ while alternating between updating $\pi_k$ and $(\tau_k, \mathbf{C}_k)$ by minimizing Eq. (8). Finally, we output the correspondences and deformed shapes and use them to compute our unsupervised loss, defined as the average alignment tightness (8).

## 3.2 Deepifying smooth shells

In order to deepify smooth shells, we need to assure that all computational steps are differentiable wrt. the inputs of the method. The update of the deformation coefficients for a fixed mapping $\mathcal{P}$ can be solved as a linear least squares problem and is therefore trivially a differentiable operation:

$$(\mathbf{X}, \mathbf{Y}) \mapsto (\mathbf{C}, \tau) := \underset{(\mathbf{C}, \tau)}{\arg \min}\, E(\mathcal{P}, \mathbf{C}, \tau). \tag{5}$$

Unfortunately, the same does not hold for the update of $\mathcal{P}$. In the discrete case, $\mathcal{P}$ is typically represented by an assignment matrix $\mathbf{P} \in \{0,1\}^{m \times n}$ with $\mathbf{P}^\top \mathbf{1} = \mathbf{1}$. In this representation, the update of the correspondences for a fixed deformation is simply a nearest neighbor search. This, however, leads to a piecewise constant energy function and therefore does not result in a differentiable mapping. In order to make the mapping from the inputs $(\mathbf{X}, \mathbf{Y})$ to the correspondences differentiable, we will replace the strict point-to-point assignment $\mathcal{P} : \mathcal{X} \to \mathcal{Y}$ with a fuzzy correspondence $\pi$:

$$\pi \in \Pi(\mathcal{X}, \mathcal{Y}). \tag{6}$$

In this context, $\Pi(\mathcal{X}, \mathcal{Y})$ is defined as the set of probability measures on the product domain $\mathcal{X} \times \mathcal{Y}$ where the marginals correspond to the surface differentials of $\mathcal{X}$ and $\mathcal{Y}$:

$$\int_{\mathcal{X}} \mathrm{d}\pi(x, y) = \mathrm{d}y\,, \quad \int_{\mathcal{Y}} \mathrm{d}\pi(x, y) = \mathrm{d}x. \tag{7}$$

Following the idea proposed in [8], we can then reformulate our matching energy from Eq. (4) using entropy regularized optimal transport (OT):

$$E(\pi, \mathbf{C}, \tau) := \int_{\mathcal{X} \times \mathcal{Y}} \left\| \mathbf{X}_k^*(x) - \mathbf{Y}_k(y) \right\|_2^2 \mathrm{d}\pi(x, y) - \lambda H(\pi). \tag{8}$$

The entropy regularization $H(\pi)$ allows us to efficiently solve for the correspondences $\pi$ using Sinkhorn's algorithm which leads to an alternating projection scheme:

$$\frac{\mathrm{d}\pi(x,y)}{\mathrm{d}x\mathrm{d}y} = \mathcal{S}^{\mathcal{X}}\big(\mathcal{S}^{\mathcal{Y}}\big(\ldots\mathcal{S}^{\mathcal{X}}\big(\mathcal{S}^{\mathcal{Y}}(p_\lambda)\big)\ldots\big)\big). \tag{9}$$

In this context, the operators $\mathcal{S}$ are projections of a given probability density $p : \mathcal{X} \times \mathcal{Y} \to \mathbb{R}$ on one of the respective constraints from Eq. (7). Moreover, the input density $p_\lambda$ is defined as:

$$p_\lambda(x,y) \propto \exp\bigg(-\frac{1}{\lambda}\big\|\mathbf{X}_k^*(x) - \mathbf{Y}_k(y)\big\|_2^2\bigg). \tag{10}$$

This scheme (9) is known to have a linear convergence [12] and we use a fixed number of iterations in practice. More importantly, each individual computation step of the resulting scheme is differentiable and [24] showed that this allows us to build Sinkhorn's algorithm into a neural network. Overall, we are able to minimize the OT energy (8) by alternating between updating $\pi_k$ and $(\tau_k, \mathbf{C}_k)$ while gradually increasing the level of detail $k$. For the correspondences $\pi_k$, this is done with the Sinkhorn scheme (9), and for the deformation coefficients $(\tau_k, \mathbf{C}_k)$, this is a weighted least squares problem.

### 3.3 Spectral convolutions

In Equation (1), we discussed how we can define a convolution filter on an input signal $F : \mathcal{X} \to \mathbb{R}^L$ in terms of an element-wise multiplication of the spectral input features $\Phi^\dagger F$. Following prior work on graph convolutions [7, 15], we use spectrally smooth filters in a low rank basis $B$:

$$G_{l'} = h\bigg(\sum_{l=1}^{L}\Phi_k\big(B\gamma_{l',l} \odot \Phi_k^\dagger F_l\big)\bigg). \tag{11}$$

Here, $\odot$ denotes the pointwise product of two vectors. Note, that we use a truncated spectral basis $\Phi_k$ instead of the full basis $\Phi$. In theory, we need infinitely many eigenfunctions $k \to \infty$ to exactly obtain a convolution with using Eq. (11). However, in the discrete world, the possible number of basis functions $\Phi_k$ is anyway bounded by the discretization coarseness and very high frequency eigenfunctions get more and more distorted by the discrete geometry. Consequently, using a moderate number of eigenpairs yields filters that are more agnostic to the discretization. Moreover, if we only use a fixed number of eigenfunctions, the approximated Fourier transform $\Phi_k^\dagger F_l$ has linear complexity. We choose the basis $B$ as a variant of the Fourier basis defined on the frequency domain:

$$B_{ij} := b_j(\lambda_i) = \cos\bigg(\frac{\lambda_i \pi j}{T}\bigg). \tag{12}$$

The Fourier basis operates on a compact or periodic domain. This is meaningful in our case because we only use a fixed number of eigenfunctions in Eq. (11). Our overall pipeline takes SHOT [40] features $F : \mathcal{X} \to \mathbb{R}^{352}$ as an input and extracts improved local features $G$ on both shapes using Equation (11). These features are then used to initialize deep shells by converting them to the initial soft correspondences $\pi$:

$$E_{\mathrm{init}}(\pi) := \int_{\mathcal{X} \times \mathcal{Y}}\big\|G^{\mathcal{X}}(x) - G^{\mathcal{Y}}(y)\big\|_2^2\mathrm{d}\pi(x,y) - \lambda H(\pi). \tag{13}$$

Our deep shells scheme then alternatingly updates the correspondences $\pi$ and the deformation coefficients $(\mathbf{C}, \tau)$ as described in the previous chapter, see Fig. 1 for an Overview. Finally, our loss function is defined as the objective value $E(\pi, \mathbf{C}, \tau)$ from Eq. (8) averaged over all deep shell iterations. Intuitively, this measures the alignment tightness of the output of the main pipeline $\mathbf{X}^*$ with the reference shape $\mathbf{Y}$ which we can compute without any supervision. Our method uses both extrinsic and intrinsic information which makes the alignment tightness a robust indicator of the matching quality without knowing the ground truth correspondences.

## 4 Results

**Implementation details** We implemented our network in PyTorch using Adam optimizer [18]. Our pipeline takes 352 dimensional SHOT descriptors [40] as an input that uses geometric features within

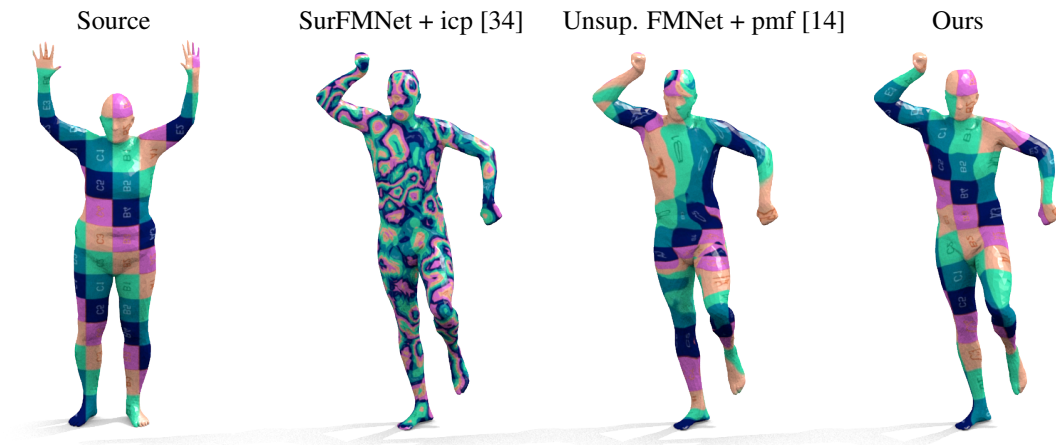

Source  SurFMNet + icp [34]  Unsup. FMNet + pmf [14]  Ours

Figure 2: A qualitative comparison corresponding to our inter-dataset experiment in the last two columns of Table 1. The source shape is from the test set of FAUST, but the target shape is from SCAPE. Our method does not require any postprocessing but still yields the best results.

5% of the shape diameter. The inputs to our method are normalized to a fixed square root area of $\frac{2}{3}$. Furthermore, we compute 500 Laplacian eigenpairs on all inputs as a preprocessing step. In particular, we use a standard cotangent discretization of the Laplace-Beltrami operator on triangular meshes with a lumped mass matrix [28]. Our spectral convolution layer uses 120 filters on the frequency domain represented with 16 cosine basis functions each, see Eq. (12). We use 200 eigenfunctions for the truncated spectral filters from Eq. (11). The high frequency Laplacian eigenfunctions on differentiable manifolds are known to grow approximately linear with a constant incline depending on the total surface area. Consequently, the 200th eigenvalue is more or less stable across surfaces which allows us to choose a fixed frequency domain from 0 to $T = 2e4$ in Eq. (12). Analogously to smooth shells [11], we use 8 iterations from $k = 6$ to $k = 20$ on a logarithmic scale for training and a refined pipeline with up to $k = 500$ eigenfunctions for testing. Finally, we use a fixed number of 10 Sinkhorn projections and the Entropy regularization coefficient $\lambda = 0.12$ in Eq. (9).

**Datasets** We evaluate our method on the standard benchmarks FAUST [4] and SCAPE [2]. Instead of the normal datasets, we use the more challenging remeshed versions from [31]. These benchmarks are known to be more realistic than the original ones. Ideally, we want correspondence methods to be agnostic to the discretization because scanning of real world objects typically leads to incompatible meshings. Therefore, the remeshed versions are an improvement over the classical FAUST and SCAPE datasets which contain templates with the same number of points and connectivity. We split both datasets into training sets of 80 and 51 shapes respectively and 20 test shapes each and randomly shuffle the $80^2$ and $51^2$ pairs during training. Although both FAUST and SCAPE contain humans, the

|  |  |  |  | *Test on - Train on* | | | |
|---|---|---|---|---|---|---|---|
|  |  | FAUST | SCAPE | F - S | S - F | F - F+S | S - F+S |
| *Axiom.* | BCICP [31] | 6.4 | 11 | - | - | - | - |
|  | ZoomOut [23] | 6.1 | 7.5 | - | - | - | - |
|  | Smooth Shells [11] | 2.5 | 4.7 | - | - | - | - |
| *Sup.* | 3D-CODED [13] | 2.5 | 31 | 33 | 31 | - | - |
|  | FMNet + pmf [20] | 11 / 5.9 | 17 / 6.3 | 33 / 14 | 30 / 11 | - | - |
|  | GeoFMNet + zo [10] | 3.1 / 1.9 | 4.4 / 3.0 | 6.0 / 4.3 | 11 / 9.2 | - | - |
| *Unsup.* | SurFMNet + icp [34] | 15 / 7.4 | 12 / 6.1. | 32 / 23 | 32 / 19 | 33 / 32 | 29 / 24 |
|  | Unsup. FMNet + pmf [14] | 10 / 5.7 | 16 / 10 | 22 / 9.3 | 29 / 12 | 11 / 6.2 | 13 / 7.7 |
|  | Ours | **1.7** | **2.5** | **2.7** | **5.4** | **1.6** | **2.4** |

Table 1: A summary of our quantitative experiments. For each result, we show the mean geodesic error in % of the shape diameter. The table is subdivided into three sections with the current state-of-the-art axiomatic, supervised and unsupervised learning approaches. The odd columns show the results on the test set of FAUST remeshed trained on FAUST remeshed, SCAPE remeshed and both datasets respectively. Analogously, the results on SCAPE are in the even columns.

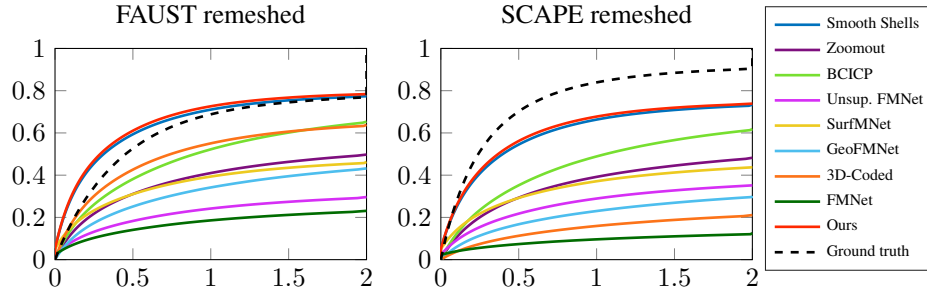

Figure 3: The cumulative conformal distortion of triangles corresponding to the first two columns of Table 1. On both datasets, our curve closely overlaps with the one from smooth shells [11]. This shows that while our method achieves a higher accuracy, it is at the same time able to obtain high quality correspondences that are comparable with smooth shells. The black dashed lines show the distortion that is caused by the remeshing of the two datasets. On FAUST remeshed, the results obtained with our method and [11] are even slightly smoother than the ground-truth.

two datasets are subject to very different challenges. FAUST contains interclass pairs of 10 different people in 10 different poses whereas the 71 SCAPE shapes all show the same person. On the other hand, the poses in SCAPE are more challenging and the geometry has less fine scale details. For example, the hands do not have discernible features like fingers which typically help disambiguate intrinsic symmetries.

**Matching accuracy**     We report the matching accuracy on the test sets in Table 1 for our method and compare it to the current state-of-the-art of both axiomatic and learning approaches. The accuracy, in this context, is defined as the mean geodesic error over all pairs and points in the dataset, normalized by the square root area $\sqrt{\text{area}(\mathcal{Y})}$. All these experiments were conducted following the Princeton benchmark protocol [17]. What is remarkable in this context is that our method outperforms the state-of-the-art, even in comparison to the best supervised methods. Moreover, our method does not require any costly postprocessing because we built a powerful matching method directly into our network. Our method also shows a quantitative improvement over the MCMC strategy from the original smooth shells pipeline [11] and it reduces the query time during testing by a large margin. See Table 2 for a runtime comparison with other methods.

**Generalization**     Aside from evaluating the matching accuracy on the individual benchmarks, we also show generalization results across different datasets, see Table 1. To that end, we apply the filters learned on FAUST remeshed to the test set of SCAPE remeshed, and vice versa. These results

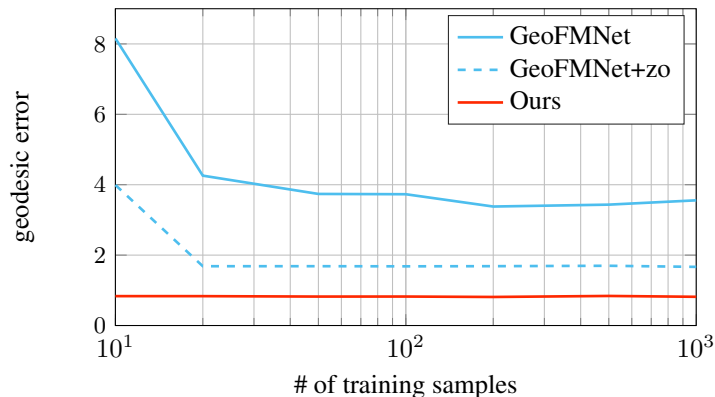

Figure 4: A quantitative comparison of our method and GeoFMNet [10] trained on SURREAL [42] with a varying number of training samples, evaluated on the test set of FAUST [4]. In particular, we show the mean geodesic error in % of the diameter with training set sizes ranging from 10 to 1000.

| B.I. [31] | ZO [23] | Sh. [11] | 3D-C. [13] | FM. [20] | G.FM. [10] | S.FM. [34] | U.FM. [14] | Ours |
|---|---|---|---|---|---|---|---|---|
| 0.9 | 1.8 | 8.7 | - | 4.9 | 1.2 | 4.9 | 4.9 | 8.7 |
| 880. | 41. | 116. | 725. | 0.2 | 0.7 | 0.8 | 0.2 | 4.9 |
| - | - | - | - | 223. | 35. | 43. | 216. | - |
| 881. | 43. | 125. | 725. | 228. | 37. | 49. | 221. | 14. |

Table 2: A runtime comparison corresponding to the experiments in the first two columns of Table 1. In the first three rows we display the average pre-processing, query and post-processing time for one pair at test time and in the last row the total time (in seconds). For training, [20] and [14] additionally require geodesic distance matrices which increases the precomputation time by $\sim 35$s per shape.

show that our learned filters are able to extract robust local features for previously unseen data, even when the local geometry of the inputs varies significantly. The final experiment we present in Table 1 shows that our method can be trained across different datasets. In particular, we train our network on all shapes of the FAUST and SCAPE training sets, including inter-dataset pairs, and test it on the individual datasets. Note, that this is only possible for an unsupervised method because there are no ground-truth labels between FAUST and SCAPE. In comparison to prior work, our method can be trained stably on this challenging setup. Remarkably, it even shows a slight improvement over the results where we trained exclusively on the individual datasets which shows that it can extract additional information from the hybrid pairs.

**Training set size** Additionally to the results on FAUST and SCAPE, we show a quantative evaluation on synthetic humans from the SURREAL dataset [42], evaluated on the test set of the FAUST [4] registrations. In comparison to GeoFMNet [10], our method produces stable results for as few as 10 training shapes, see Fig. 4. These results suggest that our robust OT matching layer is particularly useful when the amount of training data is limited.

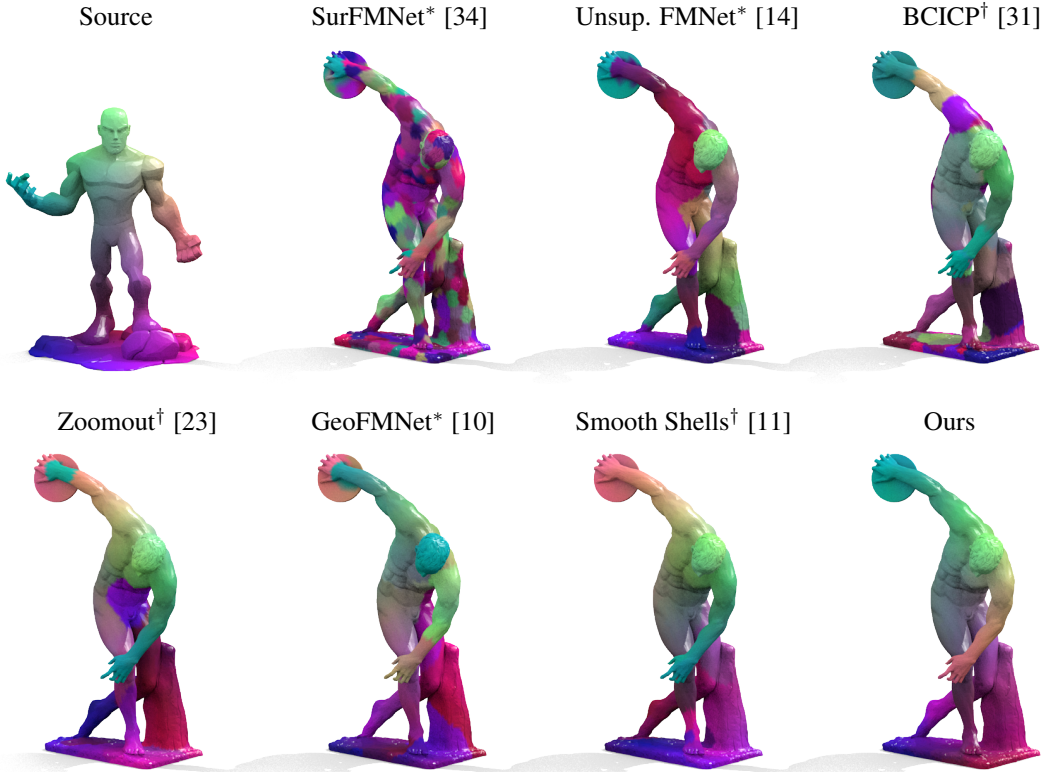

Figure 5: A qualitative comparison on a challenging non-isometric pair. The source shape is from [14], the target shape is a real scan of the statue "Discobolus" from the British Museum in London [1]. Methods with a star (*) require postprocessing and (†) denotes axiomatic approaches.

**Qualitative experiments**   The results in the last two columns of Table 1 indicate that our method is able to learn robust features that are applicable to hybrid pairs between FAUST and SCAPE. Unfortunately, we cannot verify this empirically due to the lack of ground-truth correspondences between the datasets. Instead, we show a qualitative example of an inter-dataset pair in Figure 2 with comparisons to the other two unsupervised methods from Table 1. Additionally, we show a qualitative evaluation on a non-isometric pair in Figure 5. Here, we use the weights trained on FAUST remeshed from our experiments in Table 1 for all but the axiomatic methods.

**Map smoothness**   Apart from the matching accuracy, we also evaluate the map smoothness of our obtained correspondences. This can be quantified as the conformal distortion of individual triangles, see [16, Eq. (3)] for a definition. Having smooth maps is crucial for most applications like information transfer or dense pose labeling because fine scale noise can distort the information. This effect can be visualized by mapping a texture from the surface $\mathcal{X}$ to $\mathcal{Y}$ which shows how faithfully high frequency details are preserved. In Figure 3, we show a quantitative comparison of the conformal distortion with other learning and axiomatic methods. Our results indicate that the quality of our maps is comparable to smooth shells, although we do not require an expensive preprocessing or the as-rigid-as-possible regularizer used by [11].

## 5   Conclusion

We presented deep shells, a new framework for 3D shape matching that is based on entropy regularized optimal transport. While most prior learning methods on geometric domains use either extrinsic or intrinsic information to obtain correspondences from local features, our approach operates on both domains jointly in a hierarchical pipeline. This embedding fully acknowledges the geometric nature of Riemannian manifolds: It is agnostic to the discretization while using both the extrinsic and intrinsic surface geometry. We show that this greatly increases the robustness of our network, even in an unsupervised setting. In comparison to closely related prior work, we use a spectral CNN feature extractor instead of refining hand-crafted descriptors independently for each vertex. Finally, we show quantitative results on 3D shape matching benchmarks that significantly increase the state-of-the-art. Besides the standard error on individual benchmarks, our method shows compelling generalization results across different datasets.

## Acknowledgements

We would like to thank Zorah Lähner and Florian Bernard for useful discussions. This work was supported by the Collaborative Research Center SFB-TRR 109 'Discretization in Geometry and Dynamics', the ERC Consolidator Grant "3D Reloaded", the Humboldt Foundation through the Sofja Kovalevskaja Award and the Helmholtz Association under the joint research school "Munich School for Data Science - MUDS".

## Broader Impact

With the ever increasing number of surface acquisition devices and techniques, the demand for algorithms that can process 3D data directly nowadays is higher than ever. The overarching goal is to perform recognition tasks directly on 3D sensory inputs, similarly to the way that we as humans make sense of our environment. In comparison to 2D images, which are a mere projection of the world surrounding us, geometric data is more robust to secondary effects like lighting conditions and general appearances of 3D objects. It is therefore imperative for the vision community to focus its efforts on both 2D and 3D understanding. Regarding ethical aspects, in the context of computer vision algorithms there is always the distinct possibility of dual use, e.g. for military aims. However, we believe that there is not an immediate risk of misuse associated with our algorithm.

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
