[Supplementary Material]

# Deep Shells: Unsupervised Shape Correspondence with Optimal Transport Supplementary Material

**Marvin Eisenberger**
Technical University of Munich
`marvin.eisenberger@in.tum.de`

**Aysim Toker**
Technical University of Munich
`aysim.toker@in.tum.de`

**Laura Leal-Taixé**
Technical University of Munich
`leal.taixe@tum.de`

**Daniel Cremers**
Technical University of Munich
`cremers@tum.de`

## A    Overview

In our experiments, we showed that our method outperforms prior approaches in terms of accuracy, map smoothness, runtime, and the ability to generalize to unseen data. Here, we provide further insights into the proposed architecture and additional qualitative results to give a more complete picture of Deep Shells.

Sec. B presents an ablation study that assesses the role of the different components of our method. In Sec. C we show additional qualitative comparisons. In Sec. D, we provide the cumulative matching curves corresponding to the numbers in Table 1 of the main paper. Finally, in Sec. E, we give more details on the extrinsic-intrinsic product space embedding inspired by smooth shells [4].

Figure 1: A comparison of the quality of different local features. Here, we show the assignment error averaged over the test set of FAUST remeshed. Besides being more accurate in general, our learned features are in particular able to faithfully distinguish between extremities like hands and feet.

| | Ours | i. SHOT+OT | ii. FMNet+OT | iii. SpecConv+ZO | iv. SpecConv+PMF |
|---|---|---|---|---|---|
| FAUST | 1.7 | 4.6 | 2.0 | 3.8 | 6.6 |
| SCAPE | 2.5 | 7.8 | 4.0 | 4.4 | 8.8 |

Table 1: The results of our ablation study, see Appendix B for more details.

## B   Ablation study

Deep Shells is comprised of a spectral CNN backbone and a differentiable optimal transport matching layer. Here, we want to assess how these different components contribute to our results in order to get a better understanding of our method. To that end, we replace different parts of our pipeline and report how it affects the geodesic error on FAUST [2] remeshed and SCAPE [1] remeshed, see Table 1 for a summary of the resulting accuracies. In particular, we perform the following ablations:

    i. Using SHOT descriptors [10] as inputs to our OT layer instead of learned features.

    ii. Replace the spectral convolution backbone with the 7 layer ResNet architecture from FMNet [6] and follow-up work [5, 9], train + evaluate this modified network from scratch.

    iii. Postprocessing our deep features with Zoomout [7] ...

    iv. ... and with PMF [11] instead of passing them to our OT matching layer.

Additionally, we compare the quality of our obtained features with SHOT (i) and FMNet+OT (ii). To that end, we directly compute nearest-neighbor correspondences from the raw features and show the average error on the first shape of the FAUST remeshed test set, see Figure 1. Finally, we explore how the number of eigenfunctions used during inference time affects the accuracies on FAUST remeshed reported in the main paper, see Fig. 2. On one hand, these results show that our method still achieves state-of-the-art performance for as few as 100 eigenfunctions. Nevertheless, using more high frequency information improves the results, in particular in terms of the local error, quantified by the conformal distortion of individual triangles.

Overall, our ablation study suggests that there is an intricate interplay between our feature extractor and the matching layer. The spectral convolution backbone quantifiably improves over the standard FMNet architecture which computes features independently per vertex. On the other hand, the OT matching layer proved to be an integral part of our pipeline: Even without the spectral CNN backbone, our results are still on par with prior work and postprocessing our deep features with different axiomatic methods impairs the results. Moreover, our OT layer yields the most regular correspondences with the least amount of local distortions, see Figure 3 in the main paper.

## C   Additional qualitative examples

To give a more complete picture, we show additional qualitative comparisons: For once, we show generalization results from FAUST to the KIDS [8] benchmark in Fig. 3. Although the local features

Figure 2: The accuracy of our method using a varying numbers of eigenfunctions during inference time. In particular, we compare the mean geodesic error in $\%$ of the shape diameter (left) and the mean conformal distortion (right) averaged over all pairs of the test set of FAUST remeshed.

| Source | FMNet* [6] | GeoFMNet* [3] | U. FMNet* [5] | SurFMNet* [9] | Ours |

Figure 3: A qualitative comparison with other learning based methods for examples from the KIDS [8] dataset. Methods with a star (*) require postprocessing. For all approaches, we use the weights trained on FAUST from the first column of Table 1 in the main paper.

of these two datasets are similar, KIDS has several poses with self-intersections which leads to noisy SHOT descriptors. Among the methods considered here, only GeoFMNet [3] does not rely on SHOT features as inputs but their point cloud feature extractor is not rotation-invariant. Consequently, it does not generalize well from the shapes in FAUST with mostly standing humans to the examples from KIDS which have a broader variety of poses. Moreover, in Figure 4, we present three more FAUST to SCAPE pairs analogously to the results in Figure 2 of the main paper.

## D  Matching curves

Here, we provide the cumulative matching curves corresponding to our quantitative results from Table 1 in the main paper, see Figure 5. These curves show the percentage of points below a certain error threshold, the ground truth correspondences would lead to a constant curve at 1.

## E  More details on smooth shells

In Eq. (3) of the main paper, we defined the $K + 6$ dimensional shape embedding which is the basis of our OT matching layer. Here, we provide additional details on this extrinsic-intrinsic embedding. For further details, we refer the interested reader to [4].

The product space embedding $\mathbf{X}_k$, defined in Eq. (3) of the paper, consists of three terms: The intrinsic features $\Phi_k$, the smoothed extrinsic coordinates $X_k$ and the outer normals $\mathbf{n}_k^{\mathcal{X}}$ of $X_k$. In particular, the intrinsic features are the first $k$ Laplace-Beltrami eigenfunctions on the surface $\mathcal{X}$. The smoothed extrinsic coordinates, on the other hand, are defined as follows:

$$X_k := \sum_{i=1}^{\infty} s_\sigma(k - i)\langle X, \phi_i\rangle_{L_2}\phi_i. \tag{1}$$

Here, $X : \mathcal{X} \to \mathbb{R}^3$ is the original extrinsic embedding function and $s_\sigma$ is the sigmoid function with a rescaling of the inputs by a scalar $\sigma > 0$. These embeddings $X_k : \mathcal{X} \to \mathbb{R}^3$ now constitute a family of approximations of the input geometry $X$ where $k$ controls the level of detail. For once, small values of $k$ yield a coarse approximation of $X$ and on the other hand $\lim_{k\to\infty} X_k = X$, see Figure 6. This property is now useful for our hierarchical matching layer: We can start with an alignment of coarse approximations of both input surfaces $\mathcal{X}$ and $\mathcal{Y}$ and then gradually increase the level of detail in our alternating optimization scheme, see Figure 1 of the main paper.

Figure 4: Additional qualitative comparisons corresponding to our inter-dataset experiments in the last two columns of Table 1 in the main paper. In particular, we compute a texturemap from three different FAUST shapes to target shapes from SCAPE and compare our results to [5] and [9].

Figure 5: The cumulative geodesic error curves corresponding to the comparison in Table 1 in the paper. The inter-dataset experiments in the last row are only feasible for unsupervised methods because there are no ground-truth labels between FAUST and SCAPE.

Figure 6: Examples of extrinsic shape approximations $X_k$ for different levels of detail $k$.