[Reviews · NeurIPS 2020]

Review 1

Summary and Contributions: This paper proposes a deep learning based method for unsupervised learning of shape correspondence, i.e. deep shells. The method is based on a recent work of smooth shell [9], which does not use deep learning. To turn the smooth shell method into a deep learning based method, this paper introduces the Skinhorn operation to make the assignment of correspondences differentiable. It also introduces the use of the graph neural networks to process the input Shot features for the shape correspondences. The proposed method is verified on two datasets: the FAUST and SCAPE datasets.

Strengths: - This paper proposes an interesting idea that introduces deep learning into an existing non deep learning framework, i.e. smooth shells for learning shape correspondences. - The sinkhorn operation is introduced into the framework to make the assignment operation differentiable. This is an important step for the method to become deep learning based. - The graph neural network is used to learning the relationship between the shot features, which are then used in the optimal transport layer for finding feature correspondences. - The proposed method is unsupervised. This is an advantage for the shape correspondences where ground truth labels are hard to obtain.

Weaknesses: - Shot features is used as the input to the graph neural network. The shot feature is preprocessed and not based on deep learning. This makes the whole framework not end-to-end. Would the results improve if the Shot feature is replaced with a deep learning based feature extractor? The network would also become end-to-end.

Correctness: Yes.

Clarity: Yes.

Relation to Prior Work: Yes.

Reproducibility: Yes

Additional Feedback: I maintain my original rating of accepting this paper. I like the way this paper uses optimal transport to turn the non-deep learning-based smooth shell into a deep learning-based method that can benefit from data learning. I am not so concern about the use of OT had appeared in Superglue (CVPR'2020) and in fact also another paper RPM-Net (CVPR'2020) for feature correspondences. Both works are meant for rigid point cloud registration, while this submission is on non-rigid structure. Furthermore, both CVPR works requires strong supervision, while this work is self-supervised. But please cite these papers. Additionally, they did refer to GeomFmap in the paper, and did comparisons in the result section.


Review 2

Summary and Contributions: The paper proposes a data-driven version of the recent Smooth Shell paper. It mainly redesigns the initialization step and the update of the correspondence to make them differentiable. The method runs faster and it shows better results respect to the axiomatic version.

Strengths: Simplification ========== I think the new way to address the initialization is easier and less prone to errors. Learning domain-dependent features is more robust than the MCMC. Also, this pipeline removes the need for ARAP energy. Computational timing ========== One critical aspect with the precedent Smooth Shell was his timing. It can take several minutes to align one single couple of meshes. Having a deep learning version, it permits to considerably speed up.

Weaknesses: Novelty ========== I am not convinced that the paper has enough novelty. This work converts Smooth Shell into a data-driven pipeline, restricting it to a specific domain and I am not surprised that it works better and faster then SmoothShell. Probably the main novel aspect to me is introducing the Optimal Transport to formulate the fuzzy matching in a matching pipeline, replacing the usual softmax (while it was used to retrieve permutation in [34]). I would love to read more about the regularization effect of introducing the data-driven approach and restricting the pipeline to a specific domain e.g. if there are some different behaviors in the shells alignment respect to Smooth Shell.

Correctness: To me, the paper is theoretically sound. I have some concerns about the experimental part: 1) I am not sure how to read Figure 3; the performances of DeepShells and SmoothShell are areally close in terms of continuity. Also, the curves do not saturate; does it means that they have a large error? Finally: which is the unit of measurement of X-axis? 2) It is a pity that in the paper there is no comparisons on the FAUST challenge or SHREC'19 Matching Humans with different connectivity, which are two challenging datasets; both where addressed by Smooth Shells. Furthermore, SHREC'19 would provide some intra-dataset quantitative measures. 3) I would like to have some insight into the failure cases or at least the more challenging scenarios for the method. For example: how resistant is the method to 3D misalignment (e.g. different rototranslations, scales, ...)? 4) I would like to have a few more details about competitors, e.g. how are ZoomOut and BCICP initialized? Do they also use 500 eigenfunctions?

Clarity: The paper is well written and the images are of high quality. I just find a bit difficult to follow paragraph 3.2; it looks a bit compressed, and I would suggest to extend it a little bit (e.g. further discuss the elements that substitute the original pieces of SmoothShells

Relation to Prior Work: I am satysfied by the background section. The section discuss previous methods, giving a clear taxonomy.

Reproducibility: Yes

Additional Feedback: My score is borderline: I think this work is interesting and it push further the state-of-the-art of the matching methods. However, I think there are some lacks in the experimental section, and I cannot see any clear progress from the Smooth Shells method. I am really curious to read the authors rebutal. Further questions: 1) Would be possible to have a video of the shells registration at test time? 2) Do you think that in the presence of some user-defined constraints (e.g. hand-placed landmarks) would be possible to include them into the pipeline? Do you think it would be beneficial in some cases e.g. distinguish symmetries? 3) From my understanding, the timing of several steps depends on the number of vertices of the models. Would be possible to have an insight into this (i.e. how much number of vertices impact the performances)? 4) Do you have tested your method on noisy shapes, e.g. change of topology, holes, partiality...? I think that SmoothShells can naturally work with them, while maybe DeepShells requires proper training. Is it correct? 5) I think that the data-driven part is mainly focused on the initialization process, and as shown in the ablation study it is the focal point for the matching quality. Is it correct? How far is the initialization of DeepShells from the one obtained with MCMC? Adding an ablation with "Learned Features + SmoothShells" would be interesting. 6) In the Smooth Shell paper, the authors show some inter-class results (e.g. matching a dog with a human). Since the proposed method is unsupervised, would be possible train in on a mixed datasets (e.g. all the shapes from TOSCA)? How general would be the method in that case? Rebuttal Comment ================= I thank the authors for their rebuttal and their effort to reply to my questions. After reading it carefully and discussing with other reviewers, I am convinced about the novelty of the work for the following reasons: 1) At my best knowledge, this is the first work that includes high-frequencies axiomatic method into a deep learning pipeline for matching, replacing standard FMAP framework. 2) It is performed in a hierarchal way 3) It includes the Optimal Transport, going beyond usual techniques (SoftMax and FMAP matrix difference) In this sense, my main concern has been solved and I rise my vote to acceptance. For the final manuscript version, I have some further suggestions: a) including some strongly non-isometric cases (change of topology, partiality, …); they are open questions in spectral-matching methods, and I find useful for future works to have at least a qualitative example or a clear comment about the inapplicability of the method. b) training on large datasets (e.g. thousands of SURREAL shapes), or at least provide some details on the training process already performed (e.g. memory used). An insight into the scalability of the method would be important to understand the applicability (since it requires to store hundreds of vectors for each shape). c) in the rebuttal, authors state that they use the standard setup for ZoomOut refinement (120 eigenfunctions), while in their test they use several times more (500). I would suggest comparing against ZoomOut using the same amount of frequencies. I think this would provide a better idea about the high-frequencies alignment capability of DeepShell. Otherwise, it is not completely clear if the performance improvement is due to the use of more frequencies, or effectiveness of the learning process. Finally, I would find it useful to include a “Limitation” section.


Review 3

Summary and Contributions: This paper proposes a differentiable initialization module for smooth shell based correspondence approach based on spectral convolution and OT loss. Convincing experimental results are shown on 2 of the 5 benchmark datasets.

Strengths: Spectral convolution exploits the neighborhood with OT loss in refining the local descriptors Shot and paper shows convincing results on 2 out of 5 benchmarks.

Weaknesses: 1) Limited Novelty: Learning optimal descriptors by refining shot descriptors through a differentiable loss function in an end to end learning pipeline, for computing correspondence, has been the backbone of several deep functional map work*(FMnet, UnsupFMnet, SURFMnet). This work uses the same idea with different tools, namely spectral convolution and entropic OT loss. Note that both these tools have been used separately for learning 3D surface descriptors and learning correspondence respectively. (Details in prior work section) 2) Motivation of the method and experimental validation do not match. In the introduction, the authors motivate their method by emphasizing that current deep functional map methods do not generalize well to unseen data and non-isometric pairs (Line 35). However, they ignore the experimental setup specially introduced, to test generalization of shot based methods, in GeomFmap, Donati et al. As for non-isometric pairs, there are no quantitative results shown in the paper to justify this claim. Note that there exists a Shrec16 benchmark for quantifying non-isometric matching. 3) There is a fundamental bottleneck why any method based on shot descriptors based method will underperform on generalization to unseen datasets with different triangulation as shot descriptors are local, unstable and highly dependent on triangulation. To test this, GeomFmap, Donati et al., proposed three benchmark settings with different triangulations which are ignored in this paper. Training on Surreal and testing on Faust, Shrec and Scape respectively. 4) There are several hyperparameters whose selection procedure and sensitivity is not stated at all. This is important for any unsupervised method and needs a subsection in paper. e.g. -120 filters on the frequency domain represented with 16 cosine basis functions. -500 LBO eigen functions for training and 200 for testing. - Line 186: We choose a fixed frequency domain from 0 to T = 2e4 in Eq. 12. -Line 189: we use a fixed number of 10 Sinkhorn projections and the Entropy regularization coefficient /lambda = 0.12 -Line 188: we use8 iterations from k = 6 to k = 20 on log scale. All these parameters were chosen based on test set performance without any validation set. In contrast, Surreal dataset contains thousands of shapes on which these hyperparameters can be cross validated with a separate validation set instead of tuning them on test set. 5) The paper contains overclaims or incorrect statements(See below) 6) The paper does not cite /discuss related work 1) on using OT loss for correspondence problems 2) on using convolution for learning surface descriptors(See Below)

Correctness: The claim that proposed method outperforms state-of-the-art (GEOMFmap, 3D-Coded) on multiple datasets is not fair for multiple reasons: 1) Authors only compare on 2 out of the 5 experimental settings introduced in Donati et. al. 2) Learning descriptors from raw point cloud and learning descriptors by refining shot initialization, as done in this paper, are two different tasks where former is much harder as it involves learning from scratch. Despite the hardness, advantage of learning descriptors from point cloud is evident when one tests on a dataset with different triangulation where shot based methods fail badly. These benchmarks are not tested at all in this paper. -Line 46: Compared to prior work, our method yields superior result without expensive pre-processing. The proposed algorithm takes shot descriptor as input and refines them with graph convolutions which, in principle, is no less expensive than previous work that refines shot descriptor per vertex using shared weight. Infact, the algorithm proposed here takes 500 LBO eigen functions, part of pre-processing, as input which is more expensive than even SURFMnet or GeomFmap.

Clarity: In addition to several issues described above, some terms are not clearly explained : - Line 41. We show prior approach is unstable for imperfect inputs. what are imperfect inputs? Does it mean implicitly the two datasets considered in this paper or does it mean that shot descriptors that are fed to shot based deep functional map. If it is former, both datasets, considered in this paper, are considered near isometric and not imperfect at all. If it is later, then it is a overclaim as neither GeomFmap or 3D-CODED, current state-of-the-art, rely on shot descriptors. Ablation study should be in the main paper and not supplement.

Relation to Prior Work: - Authors must cite and discuss related work on learning correspondence with OT loss especially the well known SuperGlue, CVPR 2020 that appeared on arxiv Nov, 2019. and a follow-up work that uses OT-loss for 2d-3d matching https://arxiv.org/abs/2003.06752 - Similarly, prior work on learning 3D surface descriptors with convolution network could also be cited and discussed 1) Well known and cited Dynamic graph CNN 2)https://arxiv.org/abs/2001.10472 (Siggraph 20, on arxiv since Jan, 2020) 3) MeshCNN (Siggraph 2019) 4) MeshNet (AAAI 19) -Prior work on deep functional is not purely intrinsic as stated in Introduction (Line 32-35). Donati et. al , GeomFmap, is a deep functional map that also uses extrinsic information by extracting features directly from point cloud alongwith intrinsic information.

Reproducibility: Yes

Additional Feedback: Current state of the art , 3D Coded and GeomFmap, has moved past shot based initialization methods that are sensitive to change in triangulation and thus, not generalizable at all on datasets with totally different triangulation. Therefore, any claim on outperforming state-of-the-art on generalization task is an overclaim in this paper without showing any experiments on the other 3 benchmark datasets on which shot based initialization method are shown to collapse badly.


Review 4

Summary and Contributions: This work proposes a differentiable version of Smooth Shells [9] for learning dense shape correspondences. This differentiability allows learning both mesh features as well as functional map based correspondences in an end-to-end trainable manner. Results on FAUST and SCAPE human shape datasets show that the proposed approach obtain state-of-the-art results even without making use of GT annotations during training.

Strengths: - Paper is well-written with concise introduction to different background concepts. - Optimizing the entropy-regularized optimal transport with soft-correspondences in a differentiable manner is technically interesting and novel. - Strong empirical performance on FAUST and SCAPE datasets even when compared to fully supervised techniques. - Proposed algorithm also seems faster compared to most existing techniques.

Weaknesses: - A main weakness of this work is that the experiments are only performed on human shape datasets of FAUST and SCAPE. It is important to demonstrate the generality of the approach with non-human objects or animal datasets (even if they are synthetic). - Since the approach uses spectral convolutions to estimate mesh features, are the learned features specific to human-like meshes? Can deep features learned on FAUST and SCAPE datasets generalize to meshes with completely different mesh topology?

Correctness: Seems correct.

Clarity: Yes, the paper is well written and clear.

Relation to Prior Work: Yes

Reproducibility: Yes

Additional Feedback: Having experiments on non-human datasets will make a great publication. After Rebuttal: After seeing other reviews and authors' response, I still lean towards acceptance of this work. I agree with others on the novelty of this work and I also agree that the main drawback seems to be the missing experiments on other existing datasets. It would be great if authors can include additional experiments and also discuss the limitations.

[Author Response · NeurIPS 2020]

First of all, we would like to thank the reviewers for their constructive feedback and thorough reviews. We will include detailed remarks and the references proposed by **R3** in the final version of the paper. Our method combines a spectral convolution feature extractor with a hierarchical, fully differentiable matching layer based on entropy regularized optimal transport and an unsupervised loss. As the reviewers acknowledged, our formulation leads to an increased accuracy and decreased computational cost, even in comparison with sota supervised methods.

**Novelty (R2, R3)**   Our formulation follows the line of work pioneered by FMNet [4] where the idea is to build an axiomatic matching method as a layer into a NN. Although there has been an abundance of follow-up work [6,7,8], our approach is the first to use a more elaborate matching layer than the standard functional maps (FM) proposed in [4]. Our OT matching layer uses both extrinsic and intrinsic embedding information and processes the input features in a coarse-to-fine manner. Moreover, we are the first to combine a spectral CNN feature extractor with an axiomatic matching method. In comparison to prior work, our network is more accurate and generalizes better across benchmarks.

**Comparison with GeoFMNet (R3)**   We believe that there was a fundamental misunderstanding with regards to our results in Table 1 and we would like to clarify this, as we believe that the main concerns of **R3** under 3.2, 3.3, 3.5, 4., 5. and 8. boil down to this: **R3** states that "dataset[s] with different triangulation where SHOT based methods fail badly [...] are not tested at all in this paper" and that "[in order] to test [the generalization to unseen datasets with different triangulation], GeoFMNet proposed three benchmark settings with different triangulations which are ignored in this paper". In this context, *we want to strongly emphasize that all the experiments in Table 1 are performed on the remeshed versions of FAUST and SCAPE*. These datasets indeed contain shapes with varying triangulation, therefore our results in Table 1 prove that our method is robust to this type of input noise. Moreover, we not only show results on the individual two datasets but also four more settings where we test the generalization between the different datasets. For these results, the meshing is again different and the SHOT descriptors are even less reliable due to varying local features, see Table 1. The remaining "3 of 5" experiments in [7, Fig. 3.] that **R3** frequently refers to are based on Surreal which is essentially a superset of FAUST with the same SMPL triangulation, similar local features and much more poses. For us, the additional value is marginal, e.g. the performance of GeoFMNet on Sur.-F/Sur.-S [7, Figure 3] and F-F/F-S [7, Table 1] are almost identical. We will state this point more clearly in the paper and thank the reviewer for the insight. For our experimental setup on FAUST and SCAPE remeshed we followed the standard protocol in this line of work [4,6,7,8].

**SHOT descriptors $\leftrightarrow$ PC feature extractor (R1,R3)**   To date, there are two orthogonal approaches to extract learned local features on 3D shapes: Treating the input shapes as an unordered collection of SHOT feature vectors [4,6,8] or as 3D point clouds [7,30]. We agree with the reviewers that SHOT descriptors are suboptimal since they are local, unstable and triangulation-dependent. However, for our purposes, we still prefer the former approach for multiple reasons: Most sota PC feature extractors are not invariant to rotations or near-isometries which is highly unnatural for 3D surfaces. According to [7, Appendix C] this leads to problems for humans in "bent over poses", see also Figure 2 of our Appendix. We believe the drastic improvements observed in Table 1 of our Appendix confirm the value of the proposed spectral convolution layer in aggregating information in the neighborhood of each point and thereby boosting the precision and robustness of the method such that even with a noisy local descriptor we achieve state-of-the-art performance.

**Detailed remarks (R2)**   1) Fig. 3. shows a comparison of the relative conformal distortion of triangles [40, Eq. (3)]. Indeed, our method is similar to [9] in terms of this error metric. In contrast to [9], we make use of a spectral convolution layer that drastically reduces the number of large-scale mismatches, see Fig. 4 and Table 1. The curves do not saturate because we evaluate the distortion on the remeshed datasets – even for a perfect matching some triangles get distorted. We will include the ground-truth curves in the plot for reference and thank the reviewer for the remark.
2) These two datasets are indeed very relevant for shape correspondence, however, it is FAUST re and SCAPE re which are to date widely accepted as the standard benchmarks for learning based shape matching methods for the following reasons: SHREC'19 contains only 44 shapes with severely varying poses and non-isometries to a degree that prohibits a meaningful train/test split (in GeoFMNet, the authors use the easier, remeshed version of SHREC'19 where all shapes have approx. the same resolution). For **R3**: The same holds true for SHREC'16 which only contains 25 shapes. The FAUST online challenge is at this point saturated with high-performing methods that specialize on humans, e.g. the public results from 3D coded [30] and smooth shells [9] involve using a human template which gives them an unfair advantage over true general-purpose matching methods. We suspect, that this is the reason why the most recent supervised and unsupervised methods [7] and [6] also refrained from evaluating themselves on this online challenge.
3) Our method implicitly assumes shapes with bounded distortion. This means that, like smooth shells [9], our method will fail for extremely non-isometric pairs (topological changes, partiality, ...), but this is even more true for the learning methods that are based on functional maps [4,6,7,8] which strongly favor nearly-isometric pairs. Regarding the "resistance [...] to 3D misalignment", our approach is invariant to rigid poses of the shapes and we set the scaling of the inputs to a fixed square root area of $\frac{2}{3}$.
4) We use the code from the authors' github pages [28,29] with a default number of eigenfunctions of 120.

[Meta-Review · NeurIPS 2020]

Although there was a substantial amount of discussion, most of the reviewers were excited about the new ideas in your work and their promise for improving shape correspondence. R2 has provided a detailed list of suggested changes for your final revision in their post-rebuttal comments; please address these in revising your paper for the camera-ready. Please also make sure your text clearly articulates the relationship of your work to GeomFmap (it's OK to call it "concurrent work" given the timing), and make all changes/clarifications promised in your rebuttal.